# RNA Viruses as Tools in Gene Therapy and Vaccine Development

**DOI:** 10.3390/genes10030189

**Published:** 2019-03-01

**Authors:** Kenneth Lundstrom

**Affiliations:** PanTherapeutics, Rte de Lavaux 49, CH1095 Lutry, Switzerland; lundstromkenneth@gmail.com; Tel.: +41-79-776-6351

**Keywords:** RNA viruses, gene therapy, vaccine, animal models, clinical trials, immunogenicity, protection, cancer therapy, prolonged survival

## Abstract

RNA viruses have been subjected to substantial engineering efforts to support gene therapy applications and vaccine development. Typically, retroviruses, lentiviruses, alphaviruses, flaviviruses rhabdoviruses, measles viruses, Newcastle disease viruses, and picornaviruses have been employed as expression vectors for treatment of various diseases including different types of cancers, hemophilia, and infectious diseases. Moreover, vaccination with viral vectors has evaluated immunogenicity against infectious agents and protection against challenges with pathogenic organisms. Several preclinical studies in animal models have confirmed both immune responses and protection against lethal challenges. Similarly, administration of RNA viral vectors in animals implanted with tumor xenografts resulted in tumor regression and prolonged survival, and in some cases complete tumor clearance. Based on preclinical results, clinical trials have been conducted to establish the safety of RNA virus delivery. Moreover, stem cell-based lentiviral therapy provided life-long production of factor VIII potentially generating a cure for hemophilia A. Several clinical trials on cancer patients have generated anti-tumor activity, prolonged survival, and even progression-free survival.

## 1. Introduction

The application of viral vectors in gene therapy and vaccine development dates back to the 1990s [1]. Although the early days of gene therapy were overshadowed by set-backs related to the death of a young patient treated with adenovirus vectors for a none-life threating disease [2], and the unexpected development of leukemia in retrovirus-based therapy of children with severe combined immunodeficiency (SCID) [3,4], recent development has been encouraging. Similarly, the field of vaccine development has seen substantial progress, particularly with novel engineered viral vectors targeting dendritic cells (DC), which are antigen presenting cells providing stimulation of immune responses [5]. Moreover, vectors based on self-replicating RNA viruses have allowed immunization with RNA to target infectious diseases and cancers [6].

Although progress has been achieved for both viral and non-viral vectors providing excellent possibilities for applications in gene therapy and vaccine development, the focus in this review is entirely on virus-based delivery systems. In this context, an overview is given on RNA virus-based vectors and their applications for treatment of various cancers and hemophilia, and for immunization studies aiming at providing protection against challenges with infectious agents and cancer-inducing tumor cells.

## 2. Viral Vectors

Although numerous studies have been carried out with DNA viruses such as adenovirus [7], adeno-associated virus (AAV) [8], herpes simplex virus (HSV) [9], and poxviruses [10], the focus here is solely on RNA viruses. Despite the common factor of harboring an RNA genome, there are significant differences between RNA viruses, which are described in Table 1. Moreover, a short description of each viral vector system is presented below.

### 2.1. Retroviruses

Retroviruses possess an ssRNA genome with an envelope structure [11]. Among retroviruses, Moloney murine leukemia virus (MMLV) has been engineered for efficient stable chromosomal integration and expression of heterologous genes [12]. MMLV transduction efficiency is more than 90% in dividing cells. A crucial part of retrovirus expression systems has been the design of packaging cell lines [13]. In this context, the stably integrated viral gag, pol, and env genes in packaging cell lines provide the means for particle formation and replication. The retroviral expression vector hosts the ψ+ packaging signal, the processing elements, and the genes of interest up to 8 kb in size introduced into a multilinker cloning site. The envelope protein determines the tropism (infectivity) of packaged particles [14]. For instance, ecotropic viruses only recognize receptors found on mouse and rat cells, whereas amphotropic viruses target receptors on a broad range of mammalian cells. Moreover, pantropic viruses are pseudotyped with a foreign envelope protein such as vesicular stomatis virus G protein (VSV-G), which also permits infection of nonmammalian cells [15]. The problematic random retroviral chromosomal integration encountered in SCID patients [3,4], demanded the engineering of vectors with targeted integration [16].

### 2.2. Lentiviruses

Although lentiviruses belong to the family of retroviruses, they possess the feature of being able to infect non-dividing cells [17,18]. Similar to conventional retroviruses, lentivirus expression vectors with associated packaging cell lines have been engineered, exemplified by recent production of lentivirus in HEK293 suspension cell cultures in bioreactors [19]. Moreover, both ecotropic [20] and pantropic [21] lentivirus vectors have been designed. The packaging capacity of inserts is similar to what has been described for conventional retroviruses. The chromosomal integration provides long-term expression. Moreover, engineering of inducible lentiviral vectors has been demonstrated by the introduction of a multiple cloning site upstream of the Tet promoter for the inducible expression of MYC by doxocycline administration in H209 and H345 small cell lung cancer cell lines [22].

### 2.3. Alphaviruses

Alphaviruses possess an ssRNA genome encapsulated in a capsid and envelope protein structure [23]. The nonstructural proteins (nsPs) expressed from alphavirus vectors form the replicon complex generating extreme RNA replication. The self-amplifying nature of the positive sense genome makes alphavirus vectors attractive for short-term transient heterologous gene expression [24]. Another feature of alphaviruses relates to the flexibility of using recombinant viral particles, RNA replicons, or layered DNA/RNA vectors for delivery. In the case of RNA and DNA-based expression, the gene of interest is introduced into the alphavirus expression vector downstream of the nsP genes, whereas viral particle production requires co-delivery of the alphavirus structural genes located on a helper vector [25]. The insert capacity of alphavirus vectors is 8 kb, but also multiple delivery with different constructs is feasible. The three alphaviruses most frequently used as delivery vehicles are based on Semliki Forest virus (SFV) [25], Sindbis virus (SIN) [26], and Venezuelan equine encephalitis virus (VEE) [27]. Moreover, the naturally occurring oncolytic M1 virus has been applied for cancer therapy [28]. One drawback of application of recombinant alphavirus particles relates to the lack of an efficient packaging system. Although packaging cell lines have been engineered for SFV and SIN [29], and XJ-160 virus [30], the titers are not compatible with high-titer virus production at large scale.

### 2.4. Flaviviruses

Similar to alphaviruses, the positive sense ssRNA flaviviruses possess self-amplifying RNA replicons [31]. Expression vector systems have been developed for Kunjin virus (KUN) for delivery of RNA, recombinant particles, and plasmid DNA. Related to the KUN expression vector, the gene of interest is inserted between the C20 core protein and the E22 envelope protein in frame with the viral polyprotein. Moreover, to facilitate virus production a packaging cell line has been engineered for KUN [32]. In addition to KUN, expression systems have been developed for West Nile virus [33,34], yellow fever virus [35,36], Dengue virus [37,38], and tick-borne encephalitis virus [39,40].

### 2.5. Rhabdoviruses

In contrast to alphaviruses and flaviviruses, rhabdoviruses carry a negative strand ssRNA genome [41]. In this context, expression vectors have been designed both for rabies virus (RABV) [42] and vesicular stomatitis virus (VSV) [43]. Due to the negative polarity of RNA, efficient transgene expression has been achieved by application of reverse genetics based on a recombinant vaccinia virus. However, insertion of the VSV N, P, and L genes downstream of a T7 promoter and an internal ribosome entry site (IRES) allowed efficient production of VSV particles in a vaccinia virus-free system [44]. Related to RABV vectors, the gene of interest is inserted between the RABV N and P genes [45]. Similar to VSV, a vaccinia virus-free system has been established for RABV particle production from cloned cDNA [46].

### 2.6. Measles Viruses

Similar to rhabdoviruses, measles viruses (MV) possess an ssRNA genome of negative polarity, which has required the design of rescue systems by reverse genetics for replicating MV from cloned DNA in a helper cell line [47,48]. MV expression vectors have been engineered with the gene of interest inserted either between the phosphoprotein (P) and the matrix protein (M) or between the hemagglutinin (HA) and the large protein (L). Recombinant MV particles can be generated by transfection of a helper cell line with recombinant MV constructs and a plasmid carrying the MV polymerase L gene. Three days after the transfection recombinant particles are harvested when the cytopathic effects reach 80%–90%.

### 2.7. Newcastle Disease Viruses

The negative-polarity ssRNA paramyxovirus Newcastle disease virus (NDV) strain LaSota has been engineered by reverse genetics into a recombinant vaccine vector [49]. NDV offers several advantages due to its host range restriction and no pre-existing immunity in the human population. High titer NDV replication can be achieved in cell lines acceptable for human vaccine development. Similarly, an NDV vector based on the Hitchner B1 strain was engineered by introducing the influenza HA gene between the P and M structural genes [50]. In an attempt to optimize foreign gene expression from NDV vectors, reverse genetics was applied to introduce the red fluorescence protein (RFP) at various sites downstream of the nucleocapsid (NP), P, M, fusion (F), hemagglutinin-neuraminidase (HN), and large polymerase (L) genes [51]. The RFP expression levels correlated with the genomic gene order of NDV 3’-NP-P-M-F-HN-L-5’ confirming the sequential transcription mechanism. This allows regulation of transgene expression by selecting the insertion site of the gene of interest. In a similar way, the thermostable avirulent NDV strain TS09-C was engineered by reverse genetics for GFP expression after gene insertion upstream of the NP, M, and L genes, revealing superior expression for the recombinant rTS-GFP/M virus [52]. One feature that has made NDV attractive for cancer therapy relates to the specific replication in tumor cells [53].

### 2.8. Picornaviruses

Coxsackievirus, a member of nonenveloped ssRNA Picornaviruses, has been developed as an expression vector system by the introduction of the renilla luciferase [54] and the interferon-γ (IFN-γ) [55], respectively, in the Coxsackievirus B3 (CVB3) strain. The CVB3-renilla was completely attenuated and demonstrated superior expression levels in mice in comparison to adenovirus-based expression [54]. Likewise, CVB3 generated high level expression of IFN-γ to prevent CVB3-induced myocarditis and to provide long-term immune responses in vivo [55]. Moreover, coxsackieviruses have been applied as oncolytic vectors in cancer therapy [56].

## 3. Preclinical Studies

Due to the large number of preclinical studies on gene therapy and vaccines conducted with RNA viruses in various animal models only some examples are presented below. Moreover, gene therapy and vaccine applications in animal studies are listed in Table 2.

### 3.1. Retroviruses

The classic approach for long-term gene expression has been to apply retrovirus vectors such as amphotropic retroviral vectors containing the mutant dihydrofolate reductase gene (DHFR) or the bacterial neomycin phosphotransferase (neo) gene for transduction of canine hemopoietic cells as a model for human gene therapy [57]. Moreover, several retroviral vectors have shown efficient transduction of undifferentiated murine embryonic and hematopoietic cells, which promoted preclinical studies in murine models [58].

In the context of cancer therapy, the yeast cytosine deaminase (CD) was expressed from the nonlytic amphotropic retroviral replicating vector (RRV) Toca 511 and evaluated in an orthotopic glioma mouse models [59]. Combination of RRV-Toca 511 and 5-fluorocytosine (5-FC) resulted in cell death and long-term survival in immune-competent mice. Moreover, the self-inactivating gammaretroviral vector SINfes.gp91 carrying the codon-optimized transgene gp91(phox) and the promoter for the X-linked form of chronic granulomatous disease (CGD) provided protection for X-CGD mice against challenges with Aspergillus fumigatus [60]. Related to adoptive cell therapy for the treatment of cancer patients, retrovirus vectors have been used for gene modifications of natural killer cells [61]. Robust expansion of gene-modified NK cells was established, which will support production of GMP grade material for clinical trials.

The self-inactivating COL7A1 retrovirus vector expressing type VII collagen was applied for the transduction of fibroblasts from recessive dystrophic epidermolysis bullosa (RDEB) patients [62]. Intradermal injection of genetically corrected RDEB fibroblasts reversed the disease phenotype in a xenograft model in nude mice. A single intradermal injection of 3 × 10^6^ genetically-corrected RDEB fibroblasts in skin grafts of mice generated type VII collagen deposition, anchoring of fibril formation at the dermal-epidermal junction, and improved dermal-epidermal adherence, paving the way for the treatment of nonhealing wounds in RDEB patients. Related to Xeroderma pigmentosium, a skin disease caused by deficiency in nucleotide excision repair (NER), a retrovirus-based strategy to provide XP-C keratinocytes with the wild-type XPC gene was evaluated in isolated keratinocyte stem cells [63]. No adverse effects such as oncogenic activation or clonal expansion were observed, and normal epidermal differentiation in both organotypic skin cultures and in a mouse model of human skin regeneration occurred.

### 3.2. Lentiviruses

Due to their broad host range lentiviruses have been applied for many different disease areas. For instance, lentivirus-based RNA silencing in the CNS has been targeted [64]. In this context, the mis-regulation and overexpression of α-synuclein, which results in its accumulation in neurons in Parkinson’s disease (PD), was targeted by lentivirus-based RNA interference (RNAi) both in the SH-Sy5Y human dopaminergic cell line and in vivo in neurons in rat striatum [65]. In another study, lentivirus-mediated shRNA-miR expression successfully repressed the PD-related GABA-producing enzyme glutamate decarboxylase 1 (GAD1) or GAD67 in a PD rat model [66]. Related to Alzheimer’s disease (AD) lentivirus-based RNA silencing of BACE1 attenuated amyloid precursor protein (APP) cleavage and β-amyloid production, which led to reduced neurodegeneration and behavioral deficits in an AD mouse model [67]. Moreover, lentivirus-based siRNA delivery decreased tau phosphorylation and the number of neurofibrillary tangles in an AD mouse model [68]. Related to spinal cord injury (SCI) lentivirus-based RNAi has been applied for targeting Aquaporin-4 (AQP4), a water channel protein playing a key role the pathophysiological process of SCI [69]. Lentivirus-mediated AQP4-RNAi delivery resulted in up-regulation of nerve growth factor (NGF) and accelerated motor function recovery. It was determined that the NGF up-regulation was associated with the inhibition of AQP4.

In an attempt to address the monogenic metabolic disease pyruvate kinase deficiency (PKD), hematopoietic stem cells (HSCs) were transduced with a lentivirus vector to replace the mutated pyruvate kinase isoenzymes L/R (PKLR) gene responsible for PKD [70]. Transplantation of lentivirus-transduced HSCs in myeoblated PKD mice normalized the erythroid compartment resulting in a corrected hematological phenotype and reversion of organ pathology. Analysis of the chromosomal insertion site in transplanted HSCs showed no presence of genotoxicity. Related to β-thalassemia and sickle cell disease a self-inactivating lentivirus vector (HPV569) containing the β-globin gene (β(A-T87Q)-globin was modified (BB305) for enhanced virus titers and improved transduction efficacy [71]. Sustained therapeutic efficacy was achieved in β-thalassemic mice. Moreover, secondary transplantations confirmed long-term safety showing no hematological or biochemical toxicity. These findings provided a good basis for conducting clinical trials. Related to X-linked severe combined immunodeficiency (SCID-X1) caused by mutations in the interleukin-2 receptor γ chain gene (IL2RG), a self-inactivating lentivirus vector expressing a codon-optimized human IL2RG gene restored T, B, and NK cell counts in bone marrow and peripheral blood, and normalized thymus and spleen cellularity and architecture in a mouse model [72].

In a highly promising approach, lentivirus vectors have been applied for chimeric antigen receptor T cell (CAR-T) technology in preclinical models of hematologic tumors [73]. In this context, potent antileukemia efficacy of CD123-redirected CAR-T cell therapy was evaluated by comparison of transiently active anti-CD123 mRNA-electroporated CART (RNA-CART123), T cell ablation with alemtuzumab after treatment with lentivirus-based transduction of anti-CD123-4-1BB-CD3ζ T cells (CART123), and T cell ablation with rituximab after treatment with CD20-coexpressing CART123 (CART123-CD20) [73]. All treatments provided rapid elimination of leukemia in a mouse model of acute myeloid leukemia (AML). Moreover, CAR-T cell persistence prior to ablation was required for four weeks to achieve a durable leukemia remission. The CAR-T technology has also been verified in relation to epithelial ovarian cancer (EOC) [74]. EOC is a particularly interesting model for CAR-T due to the pattern of diffusion within the peritoneal cavity, the tumor microenvironment, and the high rate of tumor associated antigens. Preclinical studies demonstrated the potential of CAR-T in EOC [74]. The CAR-T technology has also been applied for the treatment of HIV-1 infections [75]. The use of Hematopoietic Stem/Progenitor Cell (HSPC) derived T cells were redirected by a protective CD4 chimeric antigen receptor (C46CD4CAR) against simian/human immunodeficiency virus (SHIV) challenges in pigtail macaques. CAR-containing cells showed no toxicity and persisted for more than two years and generated multilineage engraftment and immune surveillance.

Finally, lentivirus vectors have been applied for delivery of shRNA to target the HIV-1 co-receptor CCR5 and the R-region of the HIV-1 long terminal repeat (LTR) as a way of treating HIV [76]. Evaluation in humanized bone marrow/thymus (hu-BLT) mice showed efficient inhibition of HIV infection, which indicates a potential alternative approach for HIV therapy. In another study, the Cal-1 anti-HIV lentiviral vector was evaluated for safety and efficacy in pigtailed macaques [77]. Robust levels of gene marking without measurable adverse events was observed in myeloid and lymphoid lineages confirming the safety of the process. Challenges with SHIV led to positive selection for gene-marked cells.

### 3.3. Alphaviruses

Self-amplifying alphavirus vectors have been subjected to numerous studies especially related to cancer gene therapy and vaccine development [78]. Local administration of replication-proficient SFV particles expressing enhanced green fluorescent protein (EGFP) prolonged survival of mice with implanted A549 lung carcinoma xenografts [79]. Moreover, an 87% reduction in tumor volume and a significantly prolonged survival of rats was observed after SFV-IL-12 administration in a syngeneic RG2 rat glioma model [80]. In another approach, six micro-RNA (miRNA) target sites were introduced into the SFV genome to target tumor cell replication [81]. Intraperitoneal administration of SFV4-miRT124 particles resulted in glioma targeting, limited spread in the CNS, and significantly prolonged survival in BALB/c mice with tumor xenografts. The naturally occurring M1 alphavirus has been evaluated in a liver tumor model, which demonstrated selective killing of zinc-finger antiviral protein (ZAP)-deficient cancer cells and potent oncolytic activity [82].

In the context of vaccine development, immunization with VEE-EBOV-NP (Ebola virus nucleoprotein) particles rendered mice resistant to challenges with lethal doses of Ebola virus [83]. Furthermore, BALB/c mice and two strains of guinea pigs were vaccinated with VEE vectors expressing Ebola glycoprotein (EBOV-GP) and NP [84]. A single administration of VEE-EBOV-GP particles or the combination of VEE-EBBOV-GP and -NP particles provided protection for both mice and guinea pigs. In contrast, mice were only protected when immunizations were carried out with VEE-EBOV-NP particles. In a recent study, it was shown that co-expression of EBOV-GP and VP40 from layered DNA/RNA SFV vectors generated both binding and neutralizing antibodies [85]. Superior immunity was achieved in comparison to a Modified Vaccinia virus Ankara (MVA) vaccine and a boost with MVA further enhanced the response. Self-amplifying alphaviruses have been further applied for vaccine development against HIV [86]. A comparison between a conventional recombinant HIV Env glycoprotein vaccine, an SFV-HIV Env RNA vector, and SFV-HIV Env particles was carried out in immunized mice [87]. The humoral immune responses were superior for the immunization with SFV particles. Moreover, SFV vectors expressing the HIV Env/Gag/polRT genes either individually or in combination were subjected to immunization experiments in mice [88]. Recombinant particles elicited stronger immune responses than RNA replicons. Immunization with VEE particles carrying the HA gene from the Hong Kong influenza A isolate (A/HK/156/97) provided protection against challenges with lethal influenza virus doses in chicken [89]. Inoculation in ovo and at one day of age resulted in partial protection, while complete protection was achieved after a single injection at two weeks of age. In another study, SFV particles carrying the influenza HA and NP genes provided protection against lethal challenges with influenza virus in immunized mice [90]. Moreover, a recent comparison of immunizations with synthetic mRNA and self-amplifying VEE RNA expressing influenza HA was made [91]. Although mice were protected by both vaccination strategies, 64 times less self-amplifying RNA (1.25 µg) compared to synthetic mRNA (80 µg) was needed. Related to Lassa virus, a bi-cistronic VEE vector with two 26S subgenomic promoters expressing Lassa virus glycoproteins of distantly clades I and IV provided protection in mice against challenges with Lassa virus [92].

### 3.4. Flaviviruses

In the context of flaviviruses, some preclinical studies have been conducted in the area of cancer therapy. For instance, non-cytopathogenic KUN particles expressing the granulocyte colony-stimulating factor (G-CSF) were intratumorally injected into mice implanted with aggressive subcutaneous CT26 colon carcinomas and B16-OVA melanomas [93]. A cure was established in over 50% of treated animals and tumor regression was associated with induced anti-cancer CD8^+^ T cells. The treatment also generated regression of CT26 lung metastases. An interesting finding was recently reported that indicated that Zika virus possesses oncolytic activity in glioblastoma stem cells (GSCs) [94]. The preferential killing of GSCs seems to be specific for Zika virus as other flaviviruses such as West Nile virus indiscriminately killed both tumor and normal neuronal cells. It was also demonstrated that Zika virus potently depleted patient-derived GSCs in culture and mice with established glioblastomas survived significantly longer in a mouse-adapted Zika virus strain. The glioblastoma targeting property of Zika virus may provide an attractive alternative for future therapeutic interventions. Related to infectious diseases, four different KUN-based simian immunodeficiency virus (SIV) vaccines (SIVmac 293 gag) were evaluated in mice [95]. The constructs included the wild-type gag gene, an RNA-optimized gag gene, a codon-optimized gag gene, and a modified gag-pol gene construct. The KUN-SIV gag-pol vaccine demonstrated the best effector memory and central memory responses, and mediation of protection.

### 3.5. Rhabdoviruses

Among applications of rhabdoviruses, recombinant VSV particles expressing the Ebola virus glycoprotein (EBOV-GP) completely protected macaques against lethal challenges with the West African EBOV-Makona strain [96]. As complete and partial protection was obtained with a single dose of VSV-EBOV-GP given as late as seven and three days before challenge, respectively, VSV-EBOV-GP may be an attractive vaccine candidate against human EBOV outbreaks. Moreover, a single injection of recombinant viral particles where EBOV and Marburg virus GPs replaced the VSV glycoprotein (G) showed complete protection of non-human primates challenged with Marburg virus and three different species of EBOV [97].

The concern of applying oncolytic VSV vectors for brain cancer relates to their inherent neurotoxicity. However, engineered pseudotyped VSV particles (VSV-GP), where the VSV-G envelope has been replaced by the non-neurotropic envelope glycoprotein from the lymphocytic choriomeningitis virus (LCMV), enhanced brain cancer cell susceptibility in vitro and were unable to infect primary human and rat neurons in vitro and in vivo, respectively [98]. Furthermore, the LCMV-pseudotyped VSV vector efficiently infected and killed all human, mouse, and canine melanoma cell lines tested as well as most human primary cultures [99]. Moreover, the survival was prolonged in both xenograft and syngeneic mouse models. Related to ovarian cancer, the pseudotyped VSV-GP vector generated oncolytic activity in ovarian cancer cell lines and in vivo [100]. The response in both subcutaneous and orthotopic xenograft models could be enhanced by combination therapy with ruxilitinib. The pseudotyped VSV-GP also provided long-term remission in prostate cancer mouse models after intratumoral injections [101]. Furthermore, intravenous administration of subcutaneous tumors and bone metastases resulted in remission.

### 3.6. Measles Viruses

The therapeutic applications of measles virus (MV) include immunization of MV vectors expressing the domain III of Dengue virus envelope protein 2 (DV2), which elicited robust neutralizing antibody responses in MV-susceptible mice [102]. Furthermore, MV vectors displaying the domain III of DV1-4 generated both neutralizing antibody responses and provided protection against challenges with four Dengue virus serotypes in mice [103]. Also, MV-expressing hepatitis B surface antigen (HBsAg) elicited humoral response in mice and primates [104].

In the context of cancer, MV vectors expressing CD46 and signaling lymphocyte activation molecule (SLAM) combined with a single-chain antibody against epidermal growth factor receptor (EGFR) at the C terminus of HA were retargeted to EGFR- or EGFRvIII primary glioblastoma cell lines and further caused tumor regression and prolonged the survival of mice with glioblastoma xenografts [105]. In another approach, the MV Edmonston strain expressing the carcinoembryonic antigen (CEA) generated significant cytopathic effect in MDA-MB231, MCF7, and SkBr3 breast cancer cell lines [106]. Tumor growth and prolonged survival was seen after intravenous administration in BALB/c mice implanted with MDA-MB231 xenografts. Related to lung cancer, MV vectors expressing SLAM (rMV-SLAMblind) showed efficient infection of human lung cancer cell lines and in mice implanted with lung xenografts injected with rMV-SLAMblind, tumor suppression tumor mass scattering in the lungs was observed [107]. Moreover, MV-SLAMblind efficiently infected and killed nectin-4 expressing pancreatic cell lines and intratumoral administration resulted in substantial growth suppression in SCID mice implanted with KLM1 and Capran-2 xenografts [108]. In the context of prostate cancer, MV was confirmed to efficiently kill PC-3, DU-145, and LNCaP prostate cancer cell lines, CEA expressing MV vectors generated significant tumor growth delay and prolonged survival after intratumoral administration in mice with subcutaneous PC-3 xenografts [109].

### 3.7. Newcastle Disease Viruses

In the context of vaccine development, complete protection against challenges with virulent NDV was obtained in specific-pathogen-free chicken immunized with the thermostable avirulent NDV strain TS09-C containing the GFP gene [52].

Related to NDV-based preclinical studies, the NDV strain 73-T caused long-lasting complete tumor regression in athymic mice implanted with human neuroblastoma and fibrosarcoma xenografts [110]. Although a single intratumoral or intraperitoneal injection generated complete regression of IMR-32 neuroblastoma xenografts in 9 out of 12 mice, multiple administration provided superior efficacy. Moreover, a modified NDV vector containing the highly fusogenic F protein exhibited significant reduction in tumor development and prolonged survival in mice with implanted CT26 tumors [111]. NDV-F vectors expressing G-CSF, IL-2, or tumor necrosis factor (TNF) were evaluated in an immunocompetent colon carcinoma model. Intratumoral administration of NDV-F-IL-2 generated a dramatic decrease in tumor growth and the majority of treated animals showed complete and long-lasting remission. Related to melanoma, an NDV vector based on the LaSota strain expressing IL-2 and IL-15 efficiently infected tumor cells and demonstrated superior tumor growth suppression compared to the recombinant NDV (rNDV) vector [112]. However, NDF-IL-15 induced significantly higher levels of IFN-γ levels than NDV-IL-2 and the survival rate was 26.67% higher for NDV-IL-15 than -IL-2. Moreover, the anti-proliferative effect of the NDV D90 strain described in the human lung cancer cell line A549 was also evaluated in athymic mice bearing implanted lung tumors [113]. Intratumoral administration of NDV D90 and rNDV-GFP suppressed both tumor growth and body weight loss. In another study, rNDV expressing IL-2 and tumor necrosis factor-related apoptosis inducing ligand (TRAIL) significantly enhanced inherent anti-neoplastic activity by inducing apoptosis and induced proliferation of CD4^+^ and CD8^+^ in mice also providing substantial reduction in tumor development [114]. Furthermore, prolonged survival was observed in mice implanted with hepatocellular carcinomas (HCCs) and melanomas, which received the rNDV-IL-2-TRAIL vector. Similarly, the NDV Anhinga strain expressing TRAIL effectively inhibited the growth of HCC in vivo, and the completely cured animals were protected against re-challenges with the same tumor cells [115]. In attempts to tackle the neurotoxicity of VSV vectors and the environmental risk of NDV, which are contagious in birds, a hybrid VSV-NDV vector where the VSV G protein was replaced by the NDV hemagglutinin-neuraminidase (HN) and the modified hyperfusogenic fusion (F) envelope proteins was engineered [116]. The hybrid vector showed reduced neurotoxicity and was avirulent in embryonated chicken eggs. Moreover, systemic administration of rVSV-NDV showed significant prolongation of survival in immune-competent mice implanted with orthotopic hepatocellular carcinoma (HCC).

### 3.8. Picornaviruses

Coxsackievirus vectors have been frequently applied for gene therapy [56]. In this context, the Coxsackievirus B3 (CVB3) expressing the human fibroblast growth factor 2 (FGF2) showed protection from ischemic necrosis after administration into ischemic hindlimbs of mice [117].

In the area of cancer therapy, Coxsackievirus A21 (CAV21) expressing the intercellular adhesion molecule-1 (ICAM-1) and decay accelerating factor (DAF) decreased tumor burden in non-obese SCID mice bearing melanoma xenografts [118]. Efficient oncolysis and systemic spread of CAV21 resulted in regression of tumors distantly from the injection site after a single administration of CAV21-ICAM1-DAF. In another study, a single intravenous injection of CAV21-ICAM1-DAF led to significant regression of pre-established T47D and MDA-MB-231-luc breast tumor xenografts and targeting and elimination of metastases [119]. Moreover, intravenous injection of CVA21-ICAM1-DAF was combined with intraperitoneal injection of doxorubicin hydrochloride resulted in significantly enhanced tumor regression compared to either viral or drug delivery alone in mice with MDA-MB-231 xenografts [120]. Related to prostate cancer, CVA21-DAFv and Echovirus 1 (EV1) showed significant oncolysis in prostate cancer cell lines and systemic delivery of EV1 induced tumor regression in the LNCaP mouse tumor model [121]. Finally, as CVB3 has been demonstrated to cause virus-induced myocarditis an approach for vaccine development represents the immunization with recombinant CVB3 expressing IFN-γ [55]. In addition to the direct IFN-γ-mediated antiviral effects, IFN-γ expression strongly activated the immune system on a long-term basis, providing complete protection against lethal challenges with Coxsackieviruses.

## 4. Clinical Trials

Despite being a relatively young field and the set-backs that were experienced, an impressively large number of clinical trials have been conducted and an increasing number of trials are planned or in progress. A summary of trials is presented below and in Table 3.

The classic example of retrovirus-based gene therapy trials relates to the treatment of children with SCID, which provided complete cure although development of leukemia occurred in some patients [3,4]. Since then, thorough vector development related to safety and efficacy issues has revitalized the application of retroviruses for clinical trials. For instance, Toca 511 retrovirus vectors have been subjected to a clinical phase I trial in patients with recurrent high-grade glioma (HGG) [122]. The overall survival rate of 13.6 months was statistically better than for the control group. A phase II/III clinical trial on HGG patients is currently in progress [123]. Gammaretroviral vectors have been evaluated in a phase I/II trial in patients with chronic granulomatous disease (CGD) characterized by primary immunodeficiency, resulting in impaired antimicrobial activity in phagocytic cells [124]. The results indicated that despite transiently resolved bacterial and fungal infections, clonal dominance and malignant transformations compromised therapeutic efficacy.

Lentivirus-based clinical trials have been less common than those for retroviruses due to their later vector development. For instance, cystic fibrosis has recently been targeted by lentivirus vectors pseudotyped with F/HN showing efficient gene transfer to lungs [125]. In preparation for clinical trials promoter/enhancer elements were assessed in mice and human air-liquid interface cultures, transduction efficiency determined, and integration site profiles were mapped. The optimized lentivirus carried a hybrid cytosine guanine dinucleotide (CpG)-free enhancer/elongation factor 1 alpha promoter (hCEF) and the cystic fibrosis membrane conductance receptor (CFTR) as the therapeutic gene, providing 90% transduction efficiency in clinically relevant delivery settings and expression of functional CFTR. These findings support the initiation of the first-in-man trial in CF patients. Lentivirus-based gene therapy has been applied for two phase I/II studies in patients with transfusion-dependent β-thalassemia [126]. Mobilized autologous CD34^+^ cells from 22 β-thalassemia patients were transduced ex vivo with the LentiGlobin BB305 vector expressing the adult hemoglobin (HbA) gene with a T87Q amino acid substitution. The lentivirus treatment reduced or eliminated the needs for long-term red cell transfusions in 22 patients with severe β-thalassemia without any serious adverse events related to the drug. Lentiviruses have found applications in treatment of leukemia. In this context, a VSV-G pseudotyped lentivirus (pELPs 19-BB-z) vector expressing a chimeric antigen receptor with specificity for the B cell antigen CD19. A low dose of autologous chimeric antigen receptor-modified T cells (1.5 × 10^5^ cells/kg body weight) reinfused into a patient with refractory chronic lymphocytic leukemia (CLL) persisted at high levels for 6 months in the blood and bone marrow [127]. Furthermore, remission continued for 10 months. Moreover, the lentivirus-based chimeric antigen receptor-modified T (CAR-T) cells targeting CD19 therapy was evaluated in 30 children and adults with relapsed acute lymphoblastic leukemia (ALL) [128]. Moreover, the lentivirus-based CAR-T therapy was recently approved by the US FDA as CTL019 or tisagenlecleucel for refractory/relapsed ALL [129]. Stem cell-based lentivirus gene therapy applied for hemophilia A treatment demonstrated life-long production of factor VIII (FVIII) and cure of disease [130]. Moreover, lentivirus-based transduction of stem cells differentiated to adipogenic, chondrogenic, and osteoblastic cells provided high level expression of factor IX applicable for hemophilia B treatment [131]. In attempts to target PD, the lentiviral-based gene therapy approach ProSavin®, the approach was to restore local and continuous dopamine production by delivery of three enzymes in the dopamine biosynthesis pathway [132]. In a Phase I/II clinical trial, patients with advanced PD showed some improvements in motor behavior. However, higher levels of dopamine replacement required for increased benefits of the treatment might be achieved by optimizing the gene order in the ProSavin® expression cassette and by engineering fusions of two or three of the transgenes. These approaches resulted in enhanced dopamine and L-Dopa production. Furthermore, the equine infectious anemia virus (EIAV)-TCiA showed significantly improved dopamine and L-Dopa production compared to ProSavin® in human neuronal cells, demonstrating expression of all three enzymes. Next, clinical evaluation of EIAV-TCiA will be conducted in PD patients. Lentivirus vectors have also been applied for development of HIV therapy by shRNA delivery targeting CCR5 [133]. Delivery of shRNAs effectively inhibited CCR5 expression resulting in protection against HIV-1 infections in cell cultures [134]. Engineering of a self-inactivating lentivirus vector expressing the sh5 anti-HIV gene and the C46 anti-viral fusion inhibitor peptide contributed to a synergistic effect on HIV-1 inhibition [133]. Moreover, the first clinical trial for lentivirus-based RNAi (LVsh5/C46) conducted in HIV patients demonstrated safety and protected the immune system from the effects of HIV without using anti-retroviral drugs [135].

In the context of alphaviruses, VEE particles expressing the CMV gB or PP65/IE1 fusion protein were evaluated in a randomized, double-blind Phase I clinical trial in CMV seronegative individuals [136]. The well-tolerated intratumoral or subcutaneous administration showed no clinically important changes, direct IFN-γ enzyme-linked immune absorbent spot (ELISPOT) responses to CMV antigens were detected in all 40 vaccinated subjects, and neutralizing antibody and multifunctional T cell responses against all three CMV antigens were obtained. Moreover, healthy HIV-negative volunteers were subjected to subcutaneous immunization with VEE particles expressing a nonmyristoylated form of HIV Gag in Phase I trials in the United States and South Africa [137]. Although the treatment was well tolerated, only modest local immune responses with low levels of binding antibodies and T cell responses was achieved. Dendritic cell (DC)-targeting VEE particles expressing CEA were subjected to intratumoral doses of 4 × 10^7^ to 4 × 10^8^ IU recombinant particles administered every 3 weeks four times in patients with advanced cancer [138]. Repeated immunization elicited clinically relevant CEA-specific T cell and antibody responses and prolonged survival was obtained in patients with CEA-specific T cell responses. A Phase I trial on VEE particles expressing the prostate-specific membrane antigen (PSMA) was carried out in patients with castration resistant metastatic prostate cancer (CRPC) [139]. CRPC patients administered with either 0.9 × 10^7^ IU or 3.6 × 10^7^ IU showed good tolerance of both doses although the detected PSMA-specific signals were weak. Despite that neither robust immune responses nor clinical benefits were obtained, the presence of neutralizing antibodies indicated that dose optimization might improve the immunogenicity. In attempts to provide passive tumor targeting, SFV particles were encapsulated in liposomes [24]. Intravenous administration of liposome encapsulated SFV particles expressing IL-12 (LipoVIL12) in kidney carcinoma and melanoma patients showed transient 10-fold enhanced IL-12 plasma levels in a phase I trial. Moreover, the encapsulation substantially enhanced tumor targeting and prevented recognition by the host immune system after repeated administration.

Clinical trials for rhabdoviruses have mainly been comprised of immunization studies for VSV vectors targeting infectious diseases, particularly EBOV. In this context, VSV vectors expressing the glycoprotein of the EBOV Zaire strain (ZEBOV) were subjected to a placebo-controlled, double-blind, dose-escalation Phase I trial in 78 volunteers receiving three doses of 3 × 10^6^, 2 × 10^7^, or 1 × 10^8^ pfu of VSV-ZEBOV [140]. Although some adverse events such as injection site pain, fatigue, myalgia, and headache occurred the overall safety profile was good. The lowest dose elicited lower antibody titers at day 28 compared to the two higher doses. Significantly increased antibody titers were observed after a second immunization on day 28, but the effect disappeared after six months. In another placebo-controlled, randomized, dose-ranging, observer-blind Phase I trial the attenuated VSVΔG- doses with sustainable IgG titers were obtained in all 40 participants [141]. Moreover, the previously applied doses of 1–5 × 10^7^ pfu were reduced to 3 × 10^5^ pfu in the dose-finding, placebo-controlled, double-blind Geneva Phase I/II study, which improved tolerability, but decreased antibody responses [142]. The superiority of the VSV-based (VSVΔG-ZEBOV-GP) vaccine in comparison to the chimpanzee adenovirus 3 (ChAd3-EBO-Z) was demonstrated in a randomized, placebo-controlled Phase III trial in 1500 adults in Liberia [143]. Adverse events such as injection site reactions, headache, fever, and fatigue were more common in individuals receiving active vaccine in comparison to placebo. Superior antibody responses one month after immunizations were obtained for VSVΔG-ZEBOV-GP (83.7%) compared to ChAd3-EBO-Z (70.8%), and placebo (2.8%). After 12 months, VSVΔG-ZEBOV-GP immunization generated better antibody responses (79.5%) than vaccination with ChAd3-EBO-Z (63.5%) and placebo (6.8%). A Phase III open-label, cluster-randomized ring vaccination trial was conducted in 7651 suspected EBOV cases in Guinea [144]. Among the participants 4123 individuals were assigned for immediate vaccination with VSV-EBOV and 3528 persons for delayed vaccination. No cases of EBOV were detected in the group receiving the vaccine at the start of the study after ten days, whereas 16 EBOV cases were discovered in the individuals receiving delayed vaccination. Overall, a good safety profile was associated with the VSV-ZEBOV vaccine also providing promising protection against EBOV. Moreover, a Phase III randomized clinical trial was conducted in 4160 individuals in Guinea and Sierra Leone [145]. A dose of 2 × 10^7^ pfu of VSV-ZEBOV was administered at the start of the trial to 2119 individuals and 2014 persons were immunized after a delay of 21 days. Substantial EBOV protection was achieved during the follow-up period of 84 days and no new cases of EBOV were detected after day 10. In another individually controlled Phase II/III clinical trial, health care and frontline workers in the five most EBOV affected districts in Sierra Leone were subjected to a single intramuscular injection of VSV-ZEBOV which was administered at enrollment or 18–24 weeks later [146]. Because of the low case frequency as the EBOV epidemic was controlled, neither EBOV cases nor vaccine-related serious adverse events were reported.

Measles viruses have been subjected to clinical trials mainly in cancer therapy. For instance, an open-label, non-randomized, dose-escalation Phase I trial was conducted in patients with cutaneous T cell lymphomas with the unmodified MV-Edm Zagreb (MV-EZ) strain [147]. Intratumoral injections of MV-EZ on days 4 and 17 preceded by subcutaneous IFNα injections (24 and 72 h earlier) defined the maximum tolerated dose as 10^3^ TCID_50_. The MV-EZ treatment resulted in complete regression of cutaneous T cell lymphoma (CTCL) lesions in one patient and partial regression in the other patients. Moreover, a Phase I trial in patients with advanced ovarian cancer was performed by intraperitoneal injection of 10^3^–10^9^ MV-CEA showing no dose-limiting toxicity [148]. Stable disease was observed in 14 patients with a median duration of 88 days and a range of 55–277 days. Related to the applied dose, 10^7^–10^9^ TCID_50_ resulted in stable disease in all patients, whereas it was accomplished for only five out of 12 with doses between 10^3^ and 10^6^ TCID_50_. MV-CEA has also been planned for a Phase I trial in patients with recurrent glioblastoma multiforme, where escalating doses from 1 × 10^5^ to 2 × 10^7^ TCID_50_ will be administered either into the resection cavity or into recurrent tumors [149]. Until now, three patients receiving 1 × 10^5^ TCID_50_ and three other patients receiving 1 × 10^6^ TCID_50_ showed no dose-limiting toxicity. Patients with relapsed refractory myeloma have been subjected to a Phase I clinical trial of intravenous administration of oncolytic MV vectors expressing the human sodium iodide symporter (NIS) [150]. As the original dose-escalation study did not reach the maximum tolerated dose (MTD), doses of 1 × 10^10^ and 1 × 10^11^ TCID_50_ were tested resulting in a complete response persisting for nine months in one patient with the higher dose, where after an isolated relapse occurred in the skull without recurrent marrow involvement. The patient remained disease-free for an additional 19 months due to an irradiation procedure.

Newcastle disease virus have been subjected to several clinical trials in the area of cancer. For instance, expression of multiple tumor-associated antigens (TAAs) from NDV vectors has provided long-term survival in Phase II trials in ovarian, stomach, and pancreatic cancers [151]. In another study, the NDV PV101 strain was intravenously administered to 79 patients with advanced solid tumors in a Phase II trial at a low dose of 1.2 × 10^10^ pfu/m^2^ and a high dose of 1.2 × 10^11^ pfu/m^2^ [152]. Administration of the higher dose resulted in objective responses and progression-free survival ranging from four to 31 months. In contrast, a randomized double-blind Phase II/III trial in melanoma patients provided no remarkable differences between individuals vaccinated with NDV and the placebo group [153]. In a more positive outcome, 335 patients with colorectal cancer were immunized with NDV vectors in a Phase III trial, which prolonged survival and improved short-term quality of life in patients [154].

Coxsackievirus vectors have been applied for clinical trials mainly for the treatment of melanoma. In a Phase I/II trial melanoma patients treated with CVA21 demonstrated good tolerance, viral replication in tumors, and increased antitumor activity [155]. Moreover, coxsackievirus vectors have been subjected to combination therapy. In this context, the antitumor activity of CVA21 was enhanced by co-treatment of melanoma patients with immune checkpoint blockade in a Phase II trial, which led to induced immune cell filtration in the tumor environment [156]. In a Phase 1b clinical trial in melanoma patients, combination therapy of CVA21 and systemic administration of pembrolizumab showed a best overall response rate of 60% and stable disease in 27% of the patients [157].

## 5. Conclusions

In summary, RNA virus-based gene therapy and vaccine development has seen some promising progress lately. This relates to engineering of novel safer and more efficient viral expression vectors and packaging systems. In addition to the broad host range and expression capacity of RNA viruses, especially self-amplifying vectors has made them attractive as gene delivery and vaccine vectors. Recent development in RNA-based therapeutics and vaccines has further strengthen the application range. The RNA-based approach further adds flexibility to drug and vaccine production procedures particularly related to well-known characteristic antigenic drift common for epidemic viruses.

Today, numerous preclinical studies have demonstrated therapeutic efficacy of RNA viruses in animal disease models and confirmation of protection against challenges with tumors and infectious agents in immunized animals. Although fewer clinical trials have been conducted in comparison to adenoviruses, the many different RNA viruses have now been subjected to clinical trials as described above. Not surprisingly, the majority of studies has focused on cancer and infectious diseases, but also other indications such as hemophilia and neurological disorders have been targeted. Encouragingly, survival rates in cancer patients have been extended and progression-free survival and complete responses have been achieved. Promising results from EBOV vaccine development leading to protection against lethal viral challenges is of great importance for targeting global epidemics. Furthermore, the recent approval by the US FDA of the lentivirus-based CAR-T technology for the treatment of refractory/relapsed ALL is a further boost for gene therapy based on RNA viruses. It is therefore expected that future development will bring a wide range of treatment technologies and vaccines for the fight against a large spectrum of diseases.

## Figures and Tables

**Table 1 genes-10-00189-t001:** Characteristics of RNA viruses.

Virus	Genome	Insert Size	Features	Ref
Retroviruses MMSV MMLV MSCV	ssRNA positive sense	8 kb	Transduction uniquely of dividing cells Packaging cell line Long-term expression Random chromosomal integration	[11,12,13,14,15,16]
Lentiviruses HIV-1 HIV-2 EIAV	ssRNA positive sense	8 kb	Broad host range (non-dividing cells) Long-term, inducible expression Chromosomal integration Low cytotoxicity	[17,18,19,20,21,22]
Alphaviruses SFV, SIN, VEE, M1	ssRNA positive sense	8 kb	Broad host range including neurons Self-amplifying RNA replicon Extreme transient expression Low immunogenicity Lack of efficient packaging system	[23,24,25,26,27,28,29,30]
Flaviviruses KUN, West Nile, YFV, Dengue virus	ssRNA positive sense	6 kb	Relatively broad host range Self-amplifying RNA replicon Transient expression	[31,32,33,34,35,36,37,38,39,40]
Rhabdoviruses Rabies VSV	ssRNA negative sense	6 kb	Relatively broad host range Self-amplifying RNA replicon Low immunogenicity	[41,42,43,44,45,46]
Measles viruses MV-Edm	ssRNA negative sense	6 kb	Self-amplifying RNA replicon Transient expression Oncolytic strains	[47,48]
NDV	ssRNA negative sense	6 kb	Replication in tumor cells Improved oncolytic vectors	[49,50,51,52,53]
Picornaviruses Coxsackievirus	ssRNA positive sense	6 kb	Oncolytic strains	[54,55,56]

HIV, human immunodeficiency virus; KUN, Kunjin virus; MMLV, Moloney murine leukemia virus; MMSV, Moloney murine sarcoma virus; MSCV, murine stem cell virus; NDV, Newcastle disease virus; SFV, Semliki Forest virus; SIN, Sindbis virus; ssRNA, single-stranded RNA; VEE, Venezuelan equine encephalitis virus; VSV, vesicular stomatitis virus; VV, vaccinia virus; YFV, yellow fever virus.

**Table 2 genes-10-00189-t002:** Examples of preclinical studies on RNA viral-based gene therapy and vaccines.

Viral Vector	Disease	Target	Response	Ref
**Retroviruses**				
RRV/Toca511+5-FC	Glioma	CD	Prolonged survival in mice	[59]
GRV	X-CGD	SINfes.gp91s	Protection against *A. fumigatus*	[60]
RV	Cancer	NK cells	Support for GMP grade production	[61]
COL7A1	RDEB	Collagen VII	Reversed RDEB in mice	[62]
RV	XP	XPC	Skin regeneration in mouse model	[63]
**Lentiviruses**				
HIV-1	PD	RNAi	Down-regulation of α-synuclein	[65]
HIV-1	PD	GAD67	Normalized neuronal activity	[66]
HIV-1	AD	RNAi	Reduction in neurodegeneration	[67]
HIV-1	AD	siRNA	Reduction in tau phosphorylation	[68]
HIV-1	SCI	AQP4-RNAi	Accelerated motor function	[69]
HIV-1	PKD	PKLR	Corrected hematological phenotype	[70]
HIV-1	β-thalassemia	β-globin	Therapeutic efficacy, no toxicity	[71]
HIV (BB305)	SCID-X1	IL2RG	Restoration of T, B, and NK cell counts	[72]
HIV-CAR-T	AML	CD123	Rapid elimination of leukemia in mice	[73]
HIV-CAR-T	Ovarian CA	TAA	Potential ovarian cancer treatment	[74]
HIV-CAR-T	SHIV	CD46-CD4	Protection against SHIV	[75]
Cal-1 anti-HIV	SHIV	Cal-1	Safe integration	[76]
**Alphaviruses**				
SFV	Lung CA	EGFP	Prolonged survival	[79]
SFV	Glioma	IL-12	Tumor regression, prolonged survival	[80]
SFV	Glioma	miRNAs	Tumor targeting, prolonged survival	[81]
M1	Liver CA	oncolytic M1	Tumor growth inhibition	[82]
VEE	EBOV	EBOV NP	Protection against Ebola virus	[83]
VEE	EBOV	EBOV GP, NP	Protection against Ebola virus	[84]
SFV	EBOV	EBOVGP, VP40	Neutralizing antibodies	[85]
SFV	HIV	Env	Humoral response	[87]
SFV	HIV	Env/Gag/Pol	Antigen-specific immune response	[88]
VEE	Influenza	HA	Protection against influenza in chicken	[89]
SFV	Influenza	HA, NP	Protection against influenza in mice	[90]
VEE	Influenza	HA	Protection against influenza in mice	[91]
VEE	Lassa	GP	Protection against Lassa in mice	[92]
**Flaviviruses**				
KUN	Colon CA	G-CSF	Tumor regression	[93]
KUN	Melanoma	G-CSF	Tumor regression	[93]
Zika virus	Glioblastoma	Glioma cells	Specific killing of GSCs	[94]
KUN	SIV	SIV gag-pol	Protection against SIV	[95]
**Rhabdoviruses**				
VSV	EBOV	EBOV-GP	Protection against EBOV in macaques	[96]
VSV	EBOV	EBOV-GP	Protection against EBOV in primates	[97]
VSV	MARV	MARV-GP	Protection against EBOV in primates	[97]
VSV	Melanoma	VSV-GP	Prolonged survival in mice	[99]
VSV + ruxilitinib	Ovarian CA	VSV-GP	Oncolytic activity	[100]
VSV	Prostate CA	VSV-GP	Long-term remission in mice	[101]
**Measles viruses**				
MV	Dengue	Dengue DV2	Neutralizing antibodies in mice	[102]
MV	Dengue	Dengue DV1-4	Protection against Dengue in mice	[103]
MV	HBV	HBsAg	Humoral response in mice, primates	[104]
MV	Brain CA	SLAM, EGFR	Tumor regression in mice	[105]
MV	Breast CA	CEA	Prolonged survival in mice	[106]
MV	Lung CA	SLAM	Tumor suppression in mice	[107]
MV	Pancreatic CA	SLAM	Tumor suppression in mice	[108]
MV	Prostate CA	CEA	Prolonged survival in mice	[109]
**NDV**				
NDV TS09-C	NDV	NDV-GFP	Protection against NDV in chicken	[52]
NDV	Neuroblastoma	NDV 73-T	Complete tumor regression in mice	[110]
NDV-F	Colon CA	IL-2	Long-term remission in mice	[111]
NDV LaSota	Melanoma	IL-15	Suppression of tumor growth in mice	[112]
NDV D90	Lung CA	NDV D90, GFP	Suppression of tumor growth in mice	[113]
NDV	Melanoma	IL-2 + TRAIL	Prolonged survival in mice	[114]
NDV	HCC	IL-2 + TRAIL	Prolonged survival in mice	[114]
NDV Anhinga	HCC	TRAIL	Cure and protection against re-challenges	[115]
VSV-NDV	HCC	VSV-NDF	Prolonged survival in mice	[116]
**Picornaviruses**				
CVB3	IN	FGF2	Protection against ischemic necrosis	[117]
CAV21	Melanoma	ICAM1, DAF	Tumor regression, also in metastases	[118]
CAV21	Breast CA	ICAM1, DAF	Improved tumor regression	[119]
CAV21 + DH	Breast CA	ICAM1. DAF1	Tumor regression, dose dependence	[120]
CAV21, EV1	Prostate CA	DAFv	Superior tumor regression	[121]
CVB3	Coxsackievirus	CVB3-IFN-γ	Protection against Coxsackievirus	[55]

5-FC, 5-fluorocytosine; AD, Alzheimer’s disease; AML, acute myeloid leukemia; AQP4, Aquaporin-4; CA, carcinoma; CAR-T, chimeric antigen receptor T cell; CAV21, Coxsackievirus A21; CEA, carcinoembryonic antigen; CD, cytosine deaminase; CVB3, Coxsackievirus B3; DAF, decay-accelerating factor; DH, doxorubicin hydrochloride; EBOV, Ebola virus; G-CSF, Granulocyte-Colony Stimulating Factor; GP, glycoprotein; GRV, gammaretrovirus; EGFR, epidermal growth factor receptor; FGF2, fibroblast growth factor-2; GSCs, glioblastoma stem cells; HA, hemagglutinin; HBV, hepatitis B virus; HBsAg, hepatitis B surface antigen; HCC, hepatocellular carcinoma; HIV, human immunodeficiency virus; ICAM-1, intercellular adhesion molecule-1; IL2RG, interleukin-2 receptor γ gene; IL, interleukin; IN, ischemic necrosis; KUN, Kunjin virus; MARB, Marburg virus; miRNAs, micro-RNAs; MV, measles virus; NDV, Newcastle disease virus; NP, nucleoprotein; PD, Parkinson’s disease; PKD, pyruvate kinase deficiency; PKLR, pyruvate kinase isoenzymes L/R; RDEB, Recessive Dystrophic Epidermolysis Bullosa; RNAi, RNA interference; RV, retroviral vector; RRV, retroviral replicating vector; SCI, spinal cord injury; SCID, severe combined immunodeficiency; SHIV, Simian/Human Immunodeficiency Virus; SFV, Semliki Forest virus; siRNA, small interfering RNA; SLAM, signaling lymphocyte activation molecule; TAA, tumor associated antigen; TRAIL, tumor necrosis factor-related apoptosis inducing ligand; VEE, Venezuelan equine encephalitis virus; VSV, vesicular stomatitis virus; VSV-GP, VSV pseudotyped with lymphocytic choriomeningitis virus glycoprotein; X-CGD, X-linked chronic granulomatous disease; XP, Xeroderma pigmentosium.

**Table 3 genes-10-00189-t003:** Examples of Clinical Trials Conducted with RNA Viral Vectors.

Viral Vector	Disease	Phase	Response	Ref
Toca 511 RV	HGG	Phase I	Prolonged survival	[122]
Toca 511 RV	HGG	Phase II/III	Study in progress	[123]
GRV	CGD	Phase I/II	Resolved bacterial & fungal infections	[124]
LV hCEF-CT	CT	Phase 0	Established parameters for Phase I CT trial	[125]
LentiGlobin BB305	β-thalassemia	Phase I/II	Reversion of need of red-blood cell transfusions	[126]
LV pELPs 19-BB-z	CLL	Case report	CLL remission for 10 months	[127]
LV CTL019	ALL	Phase I	Sustained ALL remission, prolonged survival	[128]
LV CTL019	ALL	Approved	FDA approval for refractory/relapsed ALL	[129]
LV-FVIII	Hemophilia A	Phase 0	Potential cure, life-long production of FVIII	[130]
LV-FIX	Hemophilia B	Phase 0	High level expression of	[131]
LV ProSavin®	PD	Phase I/II	Improvements in motor behavior	[132]
LVsh5/C46	HIV	Phase I	Safe, protection of the immune system	[133]
VEE-gB/pp65	CMV	Phase I	CMV-specific antibodies	[136]
VEE-gag	HIV	Phase I	Low-level antibody responses	[137]
VEE-CEA	Breast, colorectal, pancreatic CA	Phase I	Prolonged survival	[138]
VEE-PSMA	CRPC	Phase I	Neutralizing antibodies	[139]
LipoVIL12	Melanoma, kidney CA	Phase I	Safe delivery, tumor targeting	[24]
VSV-ZEBOV	EBOV	Phase I	Safe administration, antibody responses	[140]
VSVΔG-ZEBOV	EBOV	Phase I	Safe administration, sustainable IgG responses	[141]
VSV-ZEBOV	EBOV	Phase I/II	Better tolerability, reduced antibody responses	[142]
VSVΔG-ZEBOV	EBOV	Phase III	Better immune response than for Ad vector	[143]
VSV-ZEBOV	EBOV	Phase III	Safe administration, protection against EBOV	[144]
VSV-ZEBOV	EBOV	Phase III	Safe administration, protection against EBOV	[145]
VSV-ZEBOV	EBOV	Phase II/III	Safe administration, no EBOV cases	[146]
MV-EZ	CTCL	Phase I	Regression of CTCL lesions	[147]
MV-CEA	Ovarian CA	Phase I	Stable disease	[148]
MV-CEA	Glioblastoma	Phase I	No dose-related toxicity in initial patients	[149]
MV-NIS	Myeloma	Phase I	Complete response in one patient	[150]
NDV-TAA	Prostate CA	Phase II	Prolonged survival	[151]
NDV PV701	Solid tumors	Phase II	Progression-free survival	[152]
NDV	Melanoma	Phase II/III	Failure to provide superiority to controls	[153]
NDV	Colorectal CA	Phase II	Prolonged survival	[154]
CVA21	Melanoma	Phase I/II	Good tolerance, anti-tumor activity	[155]
CVA2 1 + ICB	Melanoma	Phase II	Enhanced anti-tumor activity	[156]
CVA21 + PLMAb	Melanoma	Phase 1b	Stable disease	[157]

Ad, adenovirus; ALL, Acute lymphocytic leukemia; CA, carcinoma; CAV21, Coxsackievirus A21; CEA, carcinoembryonic antigen; CGD, chronic granulomatous disease; CMV, cytomegalovirus; CRPC, castration resistant metastatic prostate cancer; CT, cystic fibrosis; CLL, chronic lymphocytic leukemia; CTCL, cutaneous T cell lymphoma; EBOV, Ebola virus; GRV, gammaretrovirus; HGG, high-grade glioma; HIV, human immunodeficiency virus; ICL, immune checkpoint blockade; IL, interleukin; LipoVIL-12; liposome-encapsulated SFV-IL-12; LV, lentivirus; MV, measles virus; NDV, Newcastle disease virus; NIS, sodium iodide symporter; PD, Parkinson’s disease; PLMab, pembrolizumab; PSMA, prostate-specific membrane antigen; RV, retroviral vector; TAA, tumor-associated antigen; VEE, Venezuelan equine encephalitis virus; VSV, vesicular stomatitis virus; ZEBOV, Ebola virus Zaire strain.

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
