# Peer review of "RNA Viruses as Tools in Gene Therapy and Vaccine Development"

_genes, 2019, doi:10.3390/genes10030189_

Reviewer 1 Report

The paper is a comprehensive review of how RNA viruses or their genomes have been engineered for gene therapy. It starts with brief introductions to the biology of the parental viruses with emphasis on the characteristics that are relevant for their usage as vectors. Next, a collection of pre-clinical and clinical studies is presented in well-ordered tables, complemented by brief descriptions in the text. Although a number of reviews covering the topic have been published in the past few years the manuscript is a valuable database for readers interested in finding a weighted summary of relevant pre-clinical and clinical studies.

I have some minor points that should be addressed by the author:

-          A major application for retroviral vectors is adoptive T cell therapy. The author only briefly cites a few papers about CAR T cells in the clinical studies part of the paper, but pre-clinical findings were not summarized. A brief paragraph reserved to this topic in the pre-clinical and clinical chapter would contribute to improve the paper

-          There is an overwhelming number of abbreviations in the manuscript. I understand that this is a result of the conciseness of the paper, and comprehensiveness of the cited studies. Nevertheless, the author should go critically through the paper and rethink if all abbreviations are needed for understanding the manuscript.

-          There is a number of typing errors in the text. A careful proofreading is necessary.

-          Here just some typos I spotted:

o   Line 174: change “amphotropid” to “amphotrophic”

o   Line 176: change “self-activating” to “self-inactivating”

o   Line 177: gp91(phax) to gp91(phox)

-          Line 240: change “six micro-RNAs” to “six micro-RNA target sites”

Author Response

Dear Reviewer 1,

Thank you for your valuable comments and suggestion. I have revised the manuscript accordingly. Please find below my responses to the issues you addressed.

-          A major application for retroviral vectors is adoptive T cell therapy. The author only briefly cites a few papers about CAR T cells in the clinical studies part of the paper, but pre-clinical findings were not summarized. A brief paragraph reserved to this topic in the pre-clinical and clinical chapter would contribute to improve the paper

A paragraph has been added for preclinical studies applying the CAR-T technology

There is an overwhelming number of abbreviations in the manuscript. I understand that this is a result of the conciseness of the paper, and comprehensiveness of the cited studies. Nevertheless, the author should go critically through the paper and rethink if all abbreviations are needed for understanding the manuscript.

Unfortunately, there is not much you can do as abbreviations are essential. However, whenever appearing in the text or in tables, definitions are provided.

-          There is a number of typing errors in the text. A careful proofreading is necessary.

-          Here just some typos I spotted:

o   Line 174: change “amphotropid” to “amphotrophic”

o   Line 176: change “self-activating” to “self-inactivating”

o   Line 177: gp91(phax) to gp91(phox)

-          Line 240: change “six micro-RNAs” to “six micro-RNA target sites

All typos have been corrected

Reviewer 2 Report

The manuscript entitled “RNA Viruses as Tools in Gene Therapy and Vaccine Development” by Kenneth Lundström is a review that covers the use of different RNA viruses as vaccines and gene therapy vectors. The author has reviewed the most relevant literature related to preclinical and clinical use of this type of viruses. However, because the field is so broad, some of the topics are not covered in depth.

Minor comments:

- In Table 1 it is stated that there is no packaging system for alphavirus. However, in the text it is described that these vectors have been used in clinical trials. Also in this table it is stated that the packaging size for alphavirus is 8 kb (also in line 89 of the text). Although in these vectors it is possible to package transgenes larger than 4-5 kb, the efficacy of packaging is dramatically reduced, and is more commonly accepted that the packaging size is around 4 kb.

Although the manuscript is in general well written, there are a number of mistakes, some of which are listed below:

Lines 67-69. Please correct the sentence

Line 83: “generating” should be “generated”

Line 105. “ a ssRNA genomes” should be “ a ssRNA genome”

Lines 120 & 121. Correct “trasnfection”

Line 126. Correct “restiction”

Line 132. Please explain: “genes showing RFP correlating the gene order of NDV 3’-NP-P-M-F-HN-L-5’”

Line 137. Correct: “attarctive”

Line 124. Indicate polarity of NDV ssRNA genome

Line 240. The following sentence is not complete: “In another approach, introduction introduction of six micro-RNAs (miRNAs) into the SFV genome to target tumor cell replication

Line 248.  “tow” should be “two”

Line 301-302. Check the sentence starting with: “…where the VSV-G envelope has been replaced by the non-neurotropic envelope…” the verb should be corrected.

528. Correct “in al patients”

Author Response

Dear Reviewer 2,

Many thanks for your comments and suggestions. I have modified the manuscript accordingly. Please find my responses below.

However, because the field is so broad, some of the topics are not covered in depth.

It is almost impossible to cover all topics and as the reviewer did not mention any missing topics it is difficult to add anything.

- In Table 1 it is stated that there is no packaging system for alphavirus. However, in the text it is described that these vectors have been used in clinical trials.

There are packaging systems for alphaviruses, but they are not very efficient. The text in Table 1 has been modified to “Lack of efficient packaging system”. Alphavirus vectors have been approved for clinical trials without the need of packaging cell line production.

Also in this table it is stated that the packaging size for alphavirus is 8 kb (also in line 89 of the text). Although in these vectors it is possible to package transgenes larger than 4-5 kb, the efficacy of packaging is dramatically reduced, and is more commonly accepted that the packaging size is around 4 kb.

I respectfully disagree with this comment as from personal experience I have packaged up to 8 kb without no significant reduction in titers. However, with larger inserts the titer decrease is quite dramatic, which can be improved to some extent by prolongation of the in vitro transcription time.

Although the manuscript is in general well written, there are a number of mistakes, some of which are listed below:

Lines 67-69. Please correct the sentence

Line 83: “generating” should be “generated”.

Actually, “generating” is correct.

Line 105. “ a ssRNA genomes” should be “ a ssRNA genome”

Lines 120 & 121. Correct “trasnfection”

Line 126. Correct “restiction”

Line 132. Please explain: “genes showing RFP correlating the gene order of NDV 3’-NP-P-M-F-HN-L-5’”

Line 137. Correct: “attarctive”

Line 124. Indicate polarity of NDV ssRNA genome

Line 240. The following sentence is not complete: “In another approach, introduction introduction of six micro-RNAs (miRNAs) into the SFV genome to target tumor cell replication

Line 248.  “tow” should be “two”

Line 301-302. Check the sentence starting with:“…where the VSV-G envelope has been replaced by the non-neurotropic envelope…” the verb should be corrected.

528. Correct “in al patients”

All typos have been corrected.